# Detoxification and Excretion of Trichothecenes in Transgenic *Arabidopsis*
*thaliana* Expressing *Fusarium graminearum* Trichothecene 3-*O*-acetyltransferase

**DOI:** 10.3390/toxins13050320

**Published:** 2021-04-29

**Authors:** Guixia Hao, Susan McCormick, Helene Tiley, Thomas Usgaard

**Affiliations:** USDA, Agricultural Research Service, Mycotoxin Prevention and Applied Microbiology Research Unit, National Center for Agricultural Utilization Research, 1815 N. University street, Peoria, IL 61604, USA; susan.mccormick@usda.gov (S.M.); helene.tiley@usda.gov (H.T.); thomas.usgaard@usda.gov (T.U.)

**Keywords:** *Fusarium graminearum*, *Arabidopsis*, transgenic plants, trichothecene, deoxynivalenol, nivalenol, NX-3, detoxification, excretion

## Abstract

*Fusarium graminearum*, the causal agent of Fusarium head blight (FHB), produces trichothecenes including deoxynivalenol (DON), nivalenol (NIV), and 3,7,15-trihydroxy-12,13-epoxytrichothec-9-ene (NX-3). These toxins contaminate grains and cause profound health problems in humans and animals. To explore exploiting a fungal self-protection mechanism in plants, we examined the ability of *F. graminearum* trichothecene 3-*O*-acetyltransferase (*FgTri101*) to detoxify several key trichothecenes produced by *F. graminearum*: DON, 15-ADON, NX-3, and NIV. *FgTri101* was cloned from *F. graminearum* and expressed in *Arabidopsis* plants. We compared the phytotoxic effects of purified DON, NIV, and NX-3 on the root growth of transgenic *Arabidopsis* expressing *FgTri101*. Compared to wild type and GUS controls, *FgTri101* transgenic *Arabidopsis* plants displayed significantly longer root length on media containing DON and NX-3. Furthermore, we confirmed that the *FgTri101* transgenic plants acetylated DON to 3-ADON, 15-ADON to 3,15-diADON, and NX-3 to NX-2, but did not acetylate NIV. Approximately 90% of the converted toxins were excreted into the media. Our study indicates that transgenic *Arabidopsis* expressing *FgTri101* can provide plant protection by detoxifying trichothecenes and excreting the acetylated toxins out of plant cells. Characterization of plant transporters involved in trichothecene efflux will provide novel targets to reduce FHB and mycotoxin contamination in economically important plant crops.

## 1. Introduction

Mycotoxins are secondary metabolites produced by fungi and pose serious threats to human and animal health by contaminating food and feed supplies. *Fusarium* species produce a variety of trichothecenes, including T-2 toxin, 4,15-diacetoxyscirpenol (DAS), nivalenol (NIV), deoxynivalenol (DON), and its acylated derivatives 3-acetyldeoxynivalenol (3-ADON) and 15-acetyldeoxynivalenol (15-ADON). Trichothecenes produced by the fungal pathogen *F. graminearum,* a major causal agent for Fusarium head blight (FHB), contaminate wheat, barley, and other small grains. Trichothecenes have multiple inhibitory effects on eukaryotic cells, including DNA and RNA synthesis, protein translation and elongation, mitochondria translation, and cell division [1,2]. Consumption of trichothecene-contaminated products causes immunotoxicity and cytotoxicity in humans and animals. In the United States, the Food and Drug Administration has issued advisory levels for DON in cereals and associated products for human consumption (1 ppm) and animal feed (5–10 ppm).

Trichothecenes play an important role during *F. graminearum* and host interactions. DON is a critical factor for FHB spread in wheat. Mutants deficient in DON production are restricted to inoculated florets and cannot pass the rachis node to spread to adjacent spikes [3,4]. On the other hand, transgenic plants expressing genes that detoxify DON increase DON and FHB resistance. For example, transgenic *Arabidopsis* and wheat plants expressing a barley UDP-glucosyltransferase, which converts DON to the less toxic DON-3-*O*-glucoside (D3G), exhibit resistance to DON [5,6]. Transgenic wheat plants expressing Fhb7, encoding a glutathione S-transferase, both detoxify trichothecenes and enhance FHB resistance [7]. Thus, the utilization of trichothecene detoxification is a promising approach to reduce FHB and mycotoxin contamination.

The gene cluster and biosynthetic pathways of trichothecenes are well characterized in *Fusarium* species. A core *TRI* gene cluster contains 12 *TRI* genes located on chromosome 2. *TRI5*, the first gene for trichothecene biosynthesis, and *TRI6,* the regulatory gene for trichothecene biosynthesis, reside in the *TRI* gene cluster [8]. For self-protection, *Fusarium* species employ a transporter (*TRI12*) to pump toxins out of the fungal cells, and a trichothecene 3-*O*-acetyltransferase (*TRI101*) to reduce toxicity of isotrichodermol and other intermediates during the biosynthesis of trichothecenes [9]. Although *TRI101* is located outside of the *TRI* gene cluster, the expression of *TRI101* is regulated by *TRI6*. In addition, Tri101 catalyzes the conversion of toxic *Fusarium* trichothecenes, such as DON, T-2 toxin, and HT-2 toxin, into less toxic C3-acetylated products [9].

The detoxification ability of *TRI101* has been exploited to reduce trichothecenes and FHB, and studies have shown that transgenic plants expressing *TRI101* convert DON to 3-ADON and have increased tolerance to trichothecenes such as DAS or DON [10,11,12,13]. However, most of these studies expressed *F. sporotrichioides TRI101 (FsTRI101)*, and transgenic plants expressing *FsTRI101* did not significantly increase wheat and barley resistance toward *F. graminearum* [11,12]. Further studies demonstrated that the proteins encoded by *FgTri101* and its ortholog FsTri101 share about 80% identity. *FgTri101* is much more effective in converting DON to 3-ADON in comparison to FsTri101 [14]. Partly because of this reason, transgenic wheat plants expressing *FsTRI101* only displayed partial resistance to FHB caused by *F. graminearum* [11]. Therefore, it is necessary to reexamine the potential to reduce FHB and trichothecenes contamination in transgenic plants expressing *FgTri101*. In addition, three genetically distinct populations of *F. graminearum* have been characterized as primary causal agents of FHB in North America [15]: NA1, which predominantly produces 15-ADON; NA2, a more aggressive population that produces 3-ADON; and a recently identified population, referred to as NA3, which produces the trichothecene analog, 3α-acetoxy, 7,15-dihydroxy-12,13-epoxytrichothec-9-ene (NX-2). Although the three populations produce different trichothecene phenotypes (chemotypes) in liquid culture, in planta the trichothecenes are primarily found in their deacetylated form. Both 3-ADON and 15-ADON are converted to DON, and NX-2 to 3,7,15-trihydroxy-12,13-epoxytrichothec-9-ene (NX-3). The toxic effects of the NX-3 toxin in inducing cytotoxicity and intestinal inflammatory are comparable to DON [16]. Control of FHB and mycotoxin contamination remains a challenge due to a lack of completely resistant varieties, a complicated fungal infection process, and emerging fungicide-resistant strains. In order to identify a successful strategy to reduce FHB and mycotoxin contamination, it is important to determine whether *FgTri101* can effectively detoxify the novel trichothecene NX-3 as well as DON.

Since wheat transformation and regeneration are time-consuming processes, in this study, we generated transgenic *Arabidopsis* plants expressing *FgTri101*. The goal of this study was to determine whether the transgenic *Arabidopsis* plants expressing the *FgTri101* gene can increase resistance to several key trichothecenes. We also examined whether *FgTri101* transgenic plants can convert DON, 15-ADON, NX-3, and NIV to their acetylated forms. 

## 2. Results

### 2.1. Generation of Transgenic Arabidopsis Expressing FgTri101

*Arabidopsis* Col-0 was transformed with the plasmid pBinARS/plus-*FgTri101,* driven by a double CaMV35S promoter (Appendix A). A total of 20 T_0_ plants were obtained on kanamycin-containing selection media. The presence of the *FgTri101* gene was confirmed in nine independent T_0_ transgenic *Arabidopsis* plants by PCR amplification (Appendix A). The expression of *FgTri101* in these transgenic plants was determined by RT-qPCR. The transgenic plant Tri101-12 displayed the highest expression level of *FgTri101* (765-fold), followed by Tri101-17 (21-fold) and Tri101-8 (14-fold), relative to the lowest expression plant Tri101-10 that had C_t_ values of 22.8 ± 0.3 (Figure 1). Four transgenic lines, Tri101-8, 10, 12, and 17, were selected for further study.

The T_2_ seeds collected from the T_1_ transgenic plants (Tri101-8, 10, 12, and 17) were examined for segregation on a kanamycin-containing selection medium. These transgenic lines displayed no segregation for kanamycin resistance and therefore were considered homozygous. Furthermore, we determined the copy number of *FgTri101* in three transgenic homozygous lines (TRI101-8-T_3_, 12-T_3_, and 17-T_3_) and nine hemizygous T_0_ TRI101 plants by quantitative real-time PCR. The *Arabidopsis* gene 4-hydroxyphenypyruvate dioxygenase (*4HPPD*) was used as an endogenous reference. The amplification efficiency of the target gene *FgTri101* and the *Arabidopsis 4HPPD* was 91% and 101.9% respectively. Using the ratio formula calculation, one transgenic plant, TRI101-6, appeared to have two insertions using DNA from hemizygous T_0_ plants (Appendix A). We estimated a single *FgTri101* insertion was present in the rest of the *Arabidopsis* transgenic lines (Appendix A). 

### 2.2. Enhanced Root Growth of Transgenic Arabidopsis Plants Expressing FgTri101 on Media Containing Trichothecenes

Since in planta trichothecenes are primarily found in their deacetylated forms, the germination and growth of transgenic *Arabidopsis* seeds were assessed on MS media containing DON, NX-3, and NIV, respectively, to evaluate the protective effect provided by *FgTri101*. On the media containing 10 mg/L DON or NX-3, the control *Arabidopsis* Col-0 and GUS transgenic seeds germinated, but displayed small yellow cotyledons, necrotic lesions on leaves, and very short roots. In contrast, the plants expressing *FgTri101* exhibited green leaves and long shoots and roots (Figure 2 and Appendix A). Statistical analyses showed that all four *FgTri101* transgenic lines displayed a significantly longer root length compared to the *Arabidopsis* Col-0 and GUS controls (*p* < 0.001) on DON and NX-3 plates (Figure 3 and Appendix A). However, on the media containing 10 mg/L NIV, *FgTri101*, *Arabidopsis* Col-0, and GUS transgenic lines displayed similar phenotypes: green leaves and long roots (Figure 2 and Appendix A). The roots of *Arabidopsis* Col-0 and GUS transgenic plants on media containing NIV were significantly longer compared with those on the media containing DON or NX-3 (Appendix A). These results suggest that expression of *FgTri101* did not provide plant protection toward NIV, and that NIV is less phytotoxic. Taken together, our observations indicate that the expression of *FgTri101* in the transgenic *Arabidopsis* plants showed enhanced resistance to DON and NX-3, but not to NIV.

### 2.3. Acetylation and Excretion of Trichothecenes by Transgenic Arabidopsis Seedlings Expressing FgTri101

To confirm the acetylation function of *FgTri101* in transgenic *Arabidopsis*, we treated *Arabidopsis* seedlings with 50 mg/L DON, 15-ADON, NX-3, or NIV in liquid half-strength MS media. After seedlings were treated for 2 days with individual toxins, the concentrations of trichothecenes were measured in *Arabidopsis* seedlings and in the media. Our results showed that DON was efficiently converted to 3-ADON in the transgenic seedlings expressing *FgTri101*, whereas no 3-ADON was detected in wild-type Col-0 and GUS transgenic seedlings (Figure 4a). This observation confirmed that the expression of *FgTri101* in transgenic *Arabidopsis* can convert DON to 3-ADON. Interestingly, we detected 3-ADON in the media growing *FgTri101* transgenic seedlings when DON was added to the media (Figure 4b). Surprisingly, over 96% of 3-ADON was detected in the media out of the total converted 3-ADON (Figure 4c). These results suggest that *Arabidopsis* transporter/transporters can efficiently pump out 3-ADON following the *FgTri101* detoxification of DON.

When 15-ADON was added, we observed reduced 15-ADON in *FgTri101* transgenic *Arabidopsis* compared to GUS transgenic *Arabidopsis*, but we did not detect 3,15-diADON in the *FgTri101* seedlings. In a repeated experiment, 3,15-diADON was detected in only a few seedling samples from each transgenic line, suggesting the conversion of 15-ADON to 3,15-diADON (Figure 5a and Appendix A). In contrast, a large amount of 3,15-diADON (98%) was detected in the media when 15-ADON was added to *FgTri101* transgenic seedling cultures (Figure 5b). 3,15-diADON was not detected in wild-type Col-0 and GUS transgenic plants or in their growth media. These results indicate that the transgenic *Arabidopsis* seedlings expressing *FgTri101* convert 15-ADON to 3,15-ADON and pump most of it to the media.

Similarly, NX-3 was converted to NX-2 in transgenic *Arabidopsis* seedling expressing *FgTri101*, but there was no conversion in wild-type Col-0 and GUS seedlings (Figure 6a). NX-2 was excreted to the media (Figure 6b), and about 90% of NX-2 was exported to the media (Figure 6c). 

However, 3-ANIV was not detected in *FgTri101* transgenic seedlings. The added NIV was observed in *Arabidopsis* seedlings expressing GUS or Tri101 (Appendix A), suggesting that *Arabidopsis* seedlings can uptake NIV, but have no ability to convert it to 3-ANIV. This explained the growth results that transgenic *Arabidopsis* expressing *FgTri101*, wild-type Col-0, or GUS had similar phenotypes on NIV-supplemented media.

Taken together, our results suggest that *Arabidopsis* expressing *FgTri101* can detoxify DON, 15-ADON, and NX-3, and efficiently pump out acetylated trichothecenes to protect the transgenic plants. However, *Arabidopsis* expressing *FgTri101* provides no protection against NIV.

### 2.4. Transgenic Arabidopsis Expressing FgTri101 Accumulated Higher Weight with Addition of Trichothecenes 

To examine the protective effect of transgenic *Arabidopsis* expressing *FgTri101*, the total weights of *Arabidopsis* seedlings were determined after being treated with 50 mg/L DON, 15-ADON, NX-3, or NIV in liquid MS media. After treatments of 48 h with DON, 15-ADON, or NX-3, the total weights of the *FgTri101* transgenic lines were significantly higher compared to the GUS transgenic controls (Figure 7a–c). However, no significant difference was observed between the NIV-treated GUS and *FgTri101* seedlings (Figure 7d). These results suggest that *FgTri101* can provide plant protection from trichothecenes DON and NX-3, but not from NIV produced by *F. graminearum*.

### 2.5. Morphology of Transgenic Arabidopsis Plants Expressing FgTri101

To assess whether the transgenic expression of *FgTri101* affects *Arabidopsis* growth and morphology, *FgTri101* transgenic lines (Tri-8, 12, and 17) and the GUS transgenic control were germinated on MS media containing kanamycin and transferred to soil. The flowering time, rosette leaf number, plant height, and number of shoots were evaluated. All three *FgTri101* expression lines did not show a significantly different flowering time, plant height, and number of shoots compared with the GUS control (Table 1). However, the *FgTri101* line 12, which had the highest expression level of *FgTri101*, displayed significantly fewer rosette leaf numbers compared with the GUS control. Nevertheless, the expression of the *FgTri101* gene without extremely high levels did not significantly affect the overall *Arabidopsis* morphology. 

## 3. Discussion

Fusarium species employ *TRI101* encoding trichothecene 3-*O*-acetyltransferase for self-protection from trichothecene mycotoxins [9]. In this study, we generated transgenic *Arabidopsis* plants expressing *FgTri101* and demonstrated that these transgenic plants had enhanced resistance to several trichothecenes compared to *Arabidopsis* Col-0 and GUS transgenic plants. We showed that *FgTri101* transgenic plants converted DON, 15-ADON, and NX-2 to their acetylated forms. Interestingly, we observed that 3-ADON, 3,15-diADON, and NX-3 were efficiently excreted into the media from *FgTri101* transgenic plants. This study demonstrated that the overexpression of *FgTri101* provides plant protection from important mycotoxins by detoxification and excretion. Further characterization of the transporters involved in toxin efflux will provide novel targets for generating transgenic crops to enhance FHB resistance and reduce mycotoxin contamination.

Gene expression analysis revealed varying levels of *FgTri101* expression in the different transgenic *Arabidopsis* plants. Among four selected transgenic lines, the gene expression of *FgTri101* was 700-fold higher in the line Tri101-12 than Tri101-10. Although the phenotypes of the four *FgTri101* lines in root growth assays using 10 mg/L individual toxins were not significantly different, Tri101-10, which had the lowest *FgTri101* expression, showed relatively low toxin conversion rates compared to Tri101-8, 12, and 17. A good correlation was demonstrated between the levels of lactoferrin protein produced in transgenic wheat lines and the level of resistance against *F. graminearum* [17]. Further investigations are needed to generate *FgTri101* antibodies and use Western blots to quantify the levels of the *FgTri101* protein.

The transgenic *Arabidopsis* plants expressing *FgTri101* displayed enhanced resistance to trichothecenes, such as DON and NX-3, with longer roots and higher growth weights. Trichothecenes are phytotoxic and affect plant morphology, such as causing inhibition of root elongation and dwarfism [18]. Plant root growth inhibition assays are commonly used to evaluate the direct phytotoxic effects of trichothecenes. DON and other structural analogs have been shown to inhibit plant growth [19,20]. Some studies showed that 3-ADON is highly toxic to wheat [21]. However, transgenic rice expressing *FgTri101* displayed enhanced DON tolerance [13]. We showed that expression of *FgTri101* not only increased *Arabidopsis* tolerance to DON, but also enhanced plant resistance toward NX-3 that is produced by a newly identified *F. graminearum* population (NA3). More importantly, we discovered that the acetylated trichothecenes were pumped out of plant cells, which may greatly reduce their phytotoxic effects. It will be interesting to determine whether the high toxicity caused by 3-ADON in wheat is due to a lack of transporters for 3-ADON excretion.

Our study demonstrated that approximately 89–98% of converted 3-ADON, 3,15-diADON, or NX-2 by Tri101 transgenic *Arabidopsis* was excreted into the media. Since 3-ADON and 3,15-di ADON have been shown to be phytotoxic to plants [20], the enhanced root growth of *FgTri101* transgenic plants may be due to most of the acylated trichothecenes being exported out of plant cells. Plant ATP-binding cassette (ABC) transporters are involved in toxin transportation and detoxification. For self-protection, many toxic metabolites are conjugated and transported to the plant vacuole for detoxification and sequestration [22]. Studies demonstrated that wheat plants contain two vacuolar membrane-localized transporters, TaABCC3.1 and TaABCC3.2, which are induced by DON in wheat and associated with DON tolerance [23]. Although lacking direct evidence of TaABCC3.1 in DON transport and detoxification, the TaABCC3.1 transporter has been demonstrated to contribute to FHB resistance in wheat [23]. *Arabidopsis* AtABCC3, the closest homologue of TaABCC3 proteins, has been shown to localize within the vacuolar membrane (tonoplast) and transport glutathione conjugates and chlorophyll catabolites [24,25]. It is possible that AtABCC3 functions in DON sequestration. Most of the ABC transporters have been characterized to localize in the vacuolar membranes and sequester toxins. For example, PDR5 from *Saccharomyces cerevisiae* is a multidrug transporter [26] and TRI12 from *F. sporotrichioides* is a trichothecenes efflux pump for fungus self-protection [27]. In the *Arabidopsis* genome, from a total of 130 annotated ABC transporters, about 20 of them have been functionally characterized. In addition to sequestering toxins in vacuoles, there are many ABC transporters involved in detoxification and transportation of various compounds [28]. A plant plasma membrane ABC transporter, NpABC1, has been identified and is involved in the secretion of an antifungal terpenoid that plays a role in plant defense [29]. To the best of our knowledge, this is the first report showing trichothecenes excreted out of plants. Further investigations are needed to determine whether wheat and barley plants expressing *FgTri101* have the ability to excrete acetylated toxins. We plan to identify the transporter/transporters that are involved in trichothecene excretion from plant cells, which may provide a novel target to reduce FHB and mycotoxin contamination. 

*FgTri101* and GUS transgenic *Arabidopsis* displayed similar healthy root length and growth weight when NIV was added, whereas GUS transgenic plants were significantly inhibited when DON or NX-3 was added. These observations suggest NIV is less phytotoxic to plants. Similarly, our recent study showed that DON, 3-ADON, and 15-ADON, but not NIV, inhibit wheat root elongation [20]. We did not observe NIV acetylation in transgenic *Arabidopsis* plants expressing *FgTri101*. *F. graminearum* strains produce NIV and acetylated NIV (typically 4,15-diANIV and 4-ANIV) in cultures [30]. To determine if Tri101 proteins are conserved between DON-producing strains and NIV-producing strains, we compared *FgTri101* from PH-1 (used in this study for generation of *Arabidopsis* transgenic plants) and Tri101 from selected NIV strains. These Tri101 proteins share high identity and the catalytic histidine (FgH156 and 160D). We did observe that two amino acids were substituted in selected NIV strains, including A215P and L392F (Appendix A). The motif DFGWG(FgD389-G393) is conserved in Fusarium species. However, the fourth position (Fg392) was observed as a Phe (F) in FsTRi101, while it is a Leu (L) in *FgTri101* [14]. In comparison, we observed that 392F was present in all NIV-procuring Fusarium species (Appendix A). This suggests that the DFGWG motif may be important for substrate specificity. Further study is needed to determine if 392F is important for *FgTri101* to bind NIV. 

Regarding the effects of 3-ADON on the growth of cereal plants, there are conflicting results. Several studies show that 3-ADON is highly toxic to wheat at low concentrations [21]. On the other hand, studies find that 3-ADON does not inhibit rice root growth [13]. Ohsato and associates suggested that the higher toxicity of 3-ADON observed in cereal plants may be due to the instability of the C3 acetyl form in plant cells. In our study, we observed most of the 3-ADON was excreted out from plant cells, which greatly reduced 3-ADON phytotoxicity. More interestingly, while we only detected a trace amount of 3,15-diADON in a few transgenic *Arabidopsis* samples, we observed significant amounts of 3,15-diADON in the media. This suggests that either 3,15-diADON is unstable inside of plant cells, or it is quickly transported out of plant cells. We speculate that transgenic plants expressing *FgTri101* may constantly export C-3 acetylated trichothecenes out of plant cells and protect crops from the phytotoxic effect of trichothecenes. Transgenic wheat and barley plants expressing *FsTRI101* converted DON to an acetylated form, 3-ADON, but did not display effective FHB resistance [11,12]. This may be because *FsTRI101* is less efficient in the conversion of DON to 3-ADON than *FgTri101* [14]. It is also possible that wheat and barley lack transporters that excrete acetylated trichothecenes. Further investigations are needed to determine whether gene stacking of *FgTri101* and the transporters for acylated toxin excretion provide effective protection for wheat and barley toward FHB and mycotoxin contamination.

Acetylation at the C-3 position has been shown to significantly reduce the phytotoxicity of a number of trichothecenes [31,32]. Therefore, expressing fungal Tri101 transacetylase could improve plant resistance to the toxins. This approach may work with other mycotoxins, but it would require different detoxification genes. 

In conclusion, we demonstrated that transgenic *Arabidopsis* plants expressing *FgTri101* displayed resistance to trichothecenes and converted several important trichothecenes to their less toxic acylated forms. More importantly, we showed that the acylated trichothecenes were excreted into the media from transgenic *Arabidopsis* plants expressing *FgTri101*. Further identification of the transporters for trichothecene excretion will provide novel breeding targets to reduce FHB and mycotoxin contamination.

## 4. Materials and Methods

### 4.1. Construction of FgTri101 Vector and Arabidopsis Transformation

*Arabidopsis thaliana* (ecotype Columbia, Col-0) were grown in a growth chamber at 23/20 °C with a 16/8 h light/dark cycle. The pBinARS/plus vector was used for *FgTri101* vector construction [33]. *FgTri101* contains no intron and was amplified using *F. graminearum* genomic DNA with primers *FgTri101*-ORF5′ and ORF3′, including the two enzyme cut sites *Spe* I and *Xba* I (Appendix A). The PCR products of *FgTri101* were purified, digested, and ligated into pBinARS/plus treated with the same enzymes. The ligation mixture was used to transform TOP10 competent cells (Invitrogen, Carlsbad, CA, USA). The positive clone containing *FgTri101* was obtained and sequenced. The purified plasmids of pBinARS/plus-*FgTri101* driven by a double 35S (D35S) promoter were introduced into *Agrobacterium tumefaciens* strain GV2260 for *Arabidopsis* transformation. 

*Arabidopsis* transformation was performed with pBinARS/plus-*FgTri101* using floral dip methods [34]. Similarly, the pBinARS/plus vector carrying β-glucuronidase (*GUS*) was used to generate transgenic *Arabidopsis* plants as controls. Transgenic plants were selected on Murashige and Skoog (MS) media with the addition of 50 mg/L kanamycin. After about two weeks, the kanamycin-resistant plants were transferred to soil and grown in the growth chamber as described above. 

### 4.2. Molecular Analysis of Transgenic Arabidopsis

Plant genomic DNA was isolated from leaves of kanamycin-resistant plants using ZR Fungal/Bacterial DNA miniprep kit (Zymo Research, Boston, MA, USA). DNA concentration and quality were evaluated using a spectrophotometer (NanoDrop 2000, Thermofisher Scientific, Waltham, MA, USA). PCR was used to amplify the target gene using the primers D35S-F and Nos-R (Appendix A). 

Total RNA was extracted from leaves of transgenic *Arabidopsis* plant containing *FgTri101* using a Trizol reagent (Sigma-Aldrich, St. Louis, MO, USA) combined with the Ambion RNA isolation kit [35]. RNA was quantified with a spectrophotometer (Nanodrop; Thermo Fisher Scientific, Waltham, MA, USA) and treated with RQ1 RNase-free DNase (Promega Corp. Madison, WI, USA). The absence of genomic DNA contamination was verified by quantitative PCR (qPCR). The first-strand cDNA was synthesized, and reverse transcriptase qPCR (RT-qPCR) was performed on a Bio-Rad CFX96 real-time system (Bio-Rad Laboratories, Hercules, CA, USA) using primers *FgTri101*-RT-5’ and *FgTri101*-RT-3’ (Appendix A). The *Arabidopsis* gene elongation factor 1-alpha (*EF1α*) was used to normalize the values as an internal control. The gene expression level was calculated with the 2^−ΔΔCt^ method using CFX manager software (Bio-Rad), relative to the transgenic plant with the lowest expression level (Tri101-10), which was set as 1. The qPCR reactions were set up in triplicate and repeated three times with similar results.

To obtain homozygotes for further analysis, T2 seeds from transgenic lines were sown on MS media containing 50 mg/L kanamycin for a progeny test. The transgenic lines for which T2 seedlings showed no segregation were considered as homozygotes. The *FgTri101* copy number was estimated by qPCR in homozygous T2 lines Tri101-8, 12, 17 and all T_0_ transgenic lines as described [35,36]. The *Arabidopsis* gene 4-hydroxyphenypyruvate dioxygenase (*4HPPD*), which is a single-copy gene, was used as the endogenous reference gene [36]. Primer efficiency for *FgTri101* and *4HPPD* was determined using a standard curve consisting of a dilution series for each primer pair. The qPCR amplification efficiency was calculated according to the following equation: Efficiency = 10 ^(−1/slope)^ − 1. The ratio of the copy number of *TRI101* was calculated using the following equation: Ratio = (1 + Efficiency (Ct*_FgTri101_*))/(1 + Efficiency (Ct_4HPPD_)). The qPCR reactions were set up in three technical triplicates and repeated three times. Data are shown as means ± SD of three replicates.

### 4.3. Mycotoxins 

DON, 15-ADON, NIV, and NX-3 were purified at the Mycotoxin Prevention and Applied Microbiology Research Unit, USDA/ARS, Peoria, IL. DON, NX-3, and NIV were dissolved in water at 5 mg/mL. 15-ADON was dissolved in 200 µl methanol and water was added to a final concentration of 5 mg/mL. 

### 4.4. Root Growth Assays 

Although *F. graminearum* produces the acetylated trichothecenes in culture, e.g., 3-ADON, 15-ADON, and NX-2, most of these toxins are converted to their deacetylated forms, DON, NIV, and NX-3 in planta. Therefore, root growth assays were conducted on a MS medium with the addition of DON, NIV, and NX-3 in square vertical plates (Greiner bio-one North America Inc. Monroe, NC). 

T_2_ seeds of the transgenic plants expressing *GUS* or *FgTri101* (lines 8, 10, 12, and 17) were surface sterilized using 70% ethanol for 2 min, then 50% Clorox for 10 min, and rinsed three times with sterile water. The seeds were sown on MS medium containing 10 mg/L DON, NIV, or NX-3. A total of nine surface-sterilized seeds were placed on each agar plate. Four plates from each transgenic line were treated with each toxin. The plates were placed vertically in a growth chamber as described above. The root length was measured every 2 days for 12 days after germination. Root length (n = 36–40) from each transgenic line was analyzed and compared with GUS controls using JMP.

### 4.5. Conversion of Trichothecenes by Transgenic Arabidopsis Seedlings

T_2_ seeds of the transgenic plants expressing *GUS* and *FgTri101* (lines 8, 10, 12, and 17) were surface sterilized as described above. The sterilized seeds were placed on a MS medium containing 50 mg/L kanamycin. The plates were kept at 4 °C for 2 days and moved to a growth chamber at 23/20 °C in a 16/8 h light/dark cycle. After 7–8 days, about 16 plants were placed into 50 mL tubes to grow in 5 mL of half-strength MS liquid medium containing 1% sucrose. After 3–4 days, the media were removed and substituted with 5 mL of fresh medium plus 50 mg/L DON, 15-ADON, NIV, or NX-3, respectively. The samples were collected two days after toxin treatments. The seedlings were rinsed three times with 5 mL of water to remove external toxins. The rinsed water was combined with the media for toxin measurement. The fresh weigh of seedlings was measured. The seedling samples and culture media were used for toxin analysis. Each transgenic line was treated with three biological replicates for each toxin. Media containing toxins without plants served as controls. 

### 4.6. Mycotoxin Extraction and Quantification

*Arabidopsis* seedlings were extracted with 10 mL of acetonitrile:water (86:14) by shaking for 1 h at 250 rpm on an orbital shaker. The samples were centrifuged for 5 min at 4000 rpm, and 9 mL of extract was dried under a stream of nitrogen [37]. For liquid media, the toxin was extracted from 1.4 mL aliquots from 20 mL that were combined with 8.6 mL acetonitrile, and the mixture was dried under a stream of nitrogen. 

Trimethylsilyl (TMS) derivatives were prepared by adding 100 µL of a 100:1 freshly prepared mixture of N-trimethylsilylimadazole/trimethylchlorosilane (Sigma-Aldrich, St. Louis, MO, USA) to the dried extract. After 30 min, 900 µL of isooctane was added to the reaction mixture followed by 1 mL water. The organic layer was transferred to 2-mL autosampler vial for GC-MS analysis. TMS derivatives of purified DON, 3-ADON, 15-ADON, 3,15-diADON, NX-2, and NX-3 (0.3125 µg to 80 µg) were similarly prepared and used to construct standard curves for quantification.

GC-MS analyses were performed on an Agilent 7890 gas chromatograph fitted with a HP-5MS column (30 m, 0.25 mm, 0.25 µm) with splitless injection and a 5977 mass detector. The injection temperature was kept at 250 °C and the column flow rate was 1 mL min^−1^. A temperature program was used with initial column temperature of 150 °C for 1 min, and then increased to 280 °C at 30 °C min^−1^ and held for 3.5 min. Samples were analyzed in both scan and selective ion monitoring (SIM) mode. Under these conditions, the TMS ether of DON was detected at 6.14 min (512, 422, 392, 295, 259, and 235 ions), 3ADON at 6.60 min (482, 467, 392, 377, 235, 193, and 181 ions), 15ADON at 6.70 min (482, 467, 392, 235, 193, and 181 ions), 3,15-diADON at 7.23 min (437, 377, 320, 262, 233, and 193 ions), NX-3 at 6.04 min (408, 305, and 181 ions), NX-2 at 6.56 min (378, 305, 275, and 181 ions), and NIV at 6.81 (510, 482, 289, 191, and 103 ions). Mass Hunter software with a NIST11 library was used to identify additional peaks in the chromatograms.

The total excreted toxin was calculated by the following formula: 3-acetylated toxin × 20/1.4, and divided by the total acetylated toxin in the plant and media.

### 4.7. Statistical Analyses

All statistical analyses were performed using JMP software. Individual analyses are discussed in each Section 4. 

## Figures and Tables

**Figure 1 toxins-13-00320-f001:**
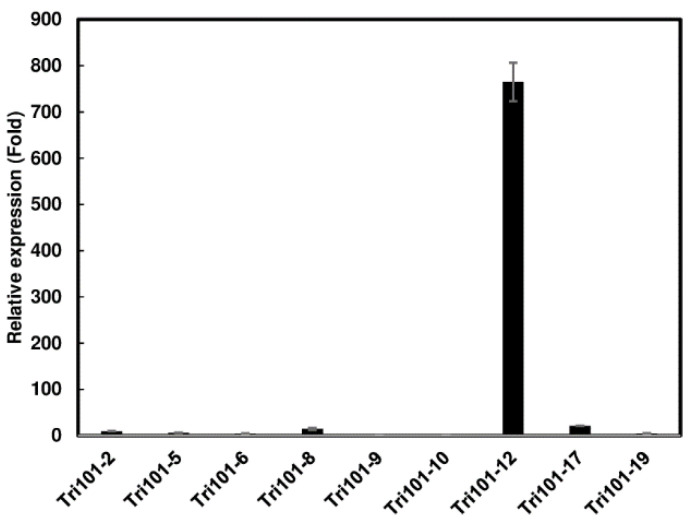
Expression of *FgTri101* in independent transgenic *Arabidopsis* plants. The expression of *FgTri101* was normalized to the expression of *Arabidopsis* elongation factor 1-alpha (*EF1α*). The relative gene expression was calculated from the 2^−ΔΔCt^ values of a sample versus *Arabidopsis FgTri101*-10, which had the lowest expression among the tested samples. The samples were run in three biological replicates with three technical replicates. Bars represent the average means from three biological replicates and their standard deviations.

**Figure 2 toxins-13-00320-f002:**
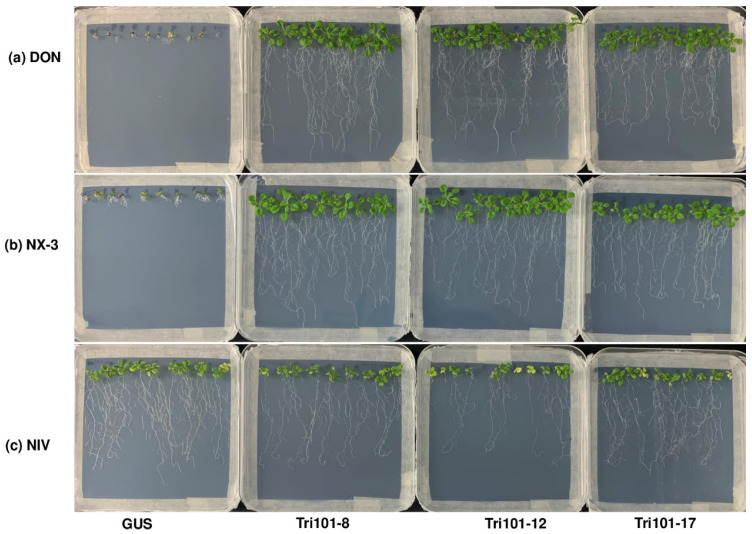
*Arabidopsis* seedling growth on MS media containing different trichothecenes (10 mg/L). (**a**) DON; (**b**) NX-3; and (**c**) NIV. Three transgenic lines (*FgTri101*-8, 12, and 17) and a control line expressing GUS were used. The photographs were taken after a two-week incubation.

**Figure 3 toxins-13-00320-f003:**
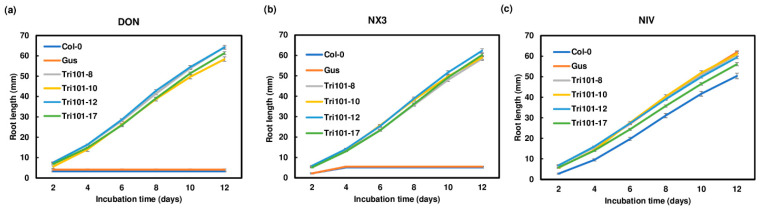
Root length comparison of the *Arabidopsis* plants expressing *FgTri101* grown on trichothecene-containing media. Seeds were germinated on half-strength MS media containing 10 mg/L of the individual toxins. (**a**) DON; (**b**) NX-3; and (**c**) NIV. Root lengths were measured every two days for two weeks. Each treatment contained 36–40 seedlings.

**Figure 4 toxins-13-00320-f004:**
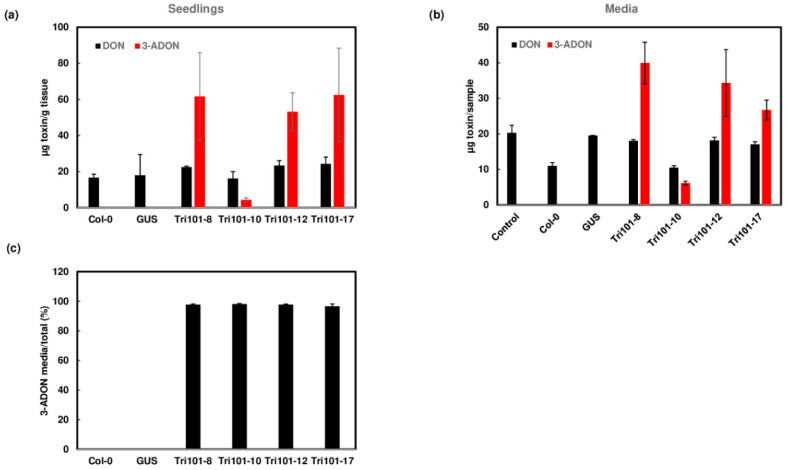
Conversion of DON to 3-ADON and excretion of 3-ADON by *FgTri101* transgenic *Arabidopsis*. (**a**) DON and 3-ADON in *Arabidopsis* seedlings after treatment with 50 mg/L DON for 2 days. All plant tissues were extracted for toxins. (**b**) DON and 3-ADON in the media after *Arabidopsis* seedlings were treated with 50 mg/L DON for 2 days. Media without plant seedlings served as a control. Toxins in 1.4 mL aliquots from 20 mL media were measured and presented. (**c**) 3-ADON in the media in comparison to total 3-ADON in the plant and media. Percentage of excreted 3-ADON: 3-ADON media/ (3-ADON media + plant) × 100.

**Figure 5 toxins-13-00320-f005:**
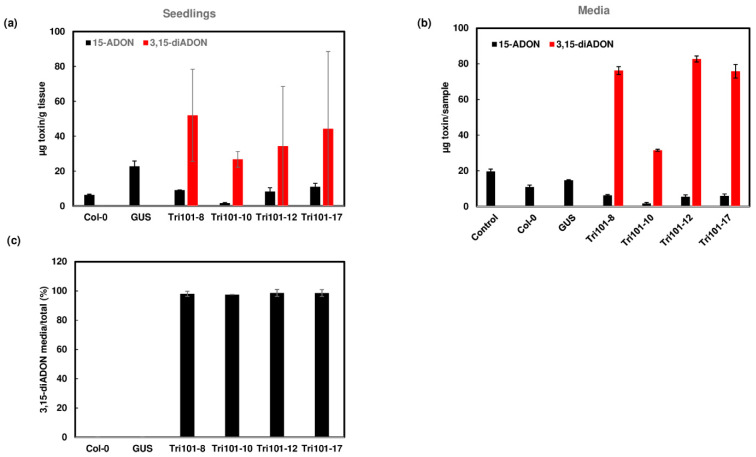
Conversion of 15-ADON to 3,15-diADON and excretion of 3,15-diADON by *FgTri101* transgenic *Arabidopsis*. (**a**) 15-ADON, and 3,15-diADON in *Arabidopsis* seedlings after treatment with 50 mg/L 15-ADON for 2 days. (**b**) 15-ADON and 3,15-diADON in the media after *Arabidopsis* seedlings were treated with 50 mg/L 15-ADON for 2 days. Media without plant seedlings served as a control. Toxins in the media were measured in 1.4 mL aliquots from 20 mL of media and presented. (**c**) 3,15-diADON in the media in comparison to 3,15-diADON in the plant and media. Percentage of excreted 3,15-diADON: 3,15-diADON media/ (3,15-diADON media + plant) × 100.

**Figure 6 toxins-13-00320-f006:**
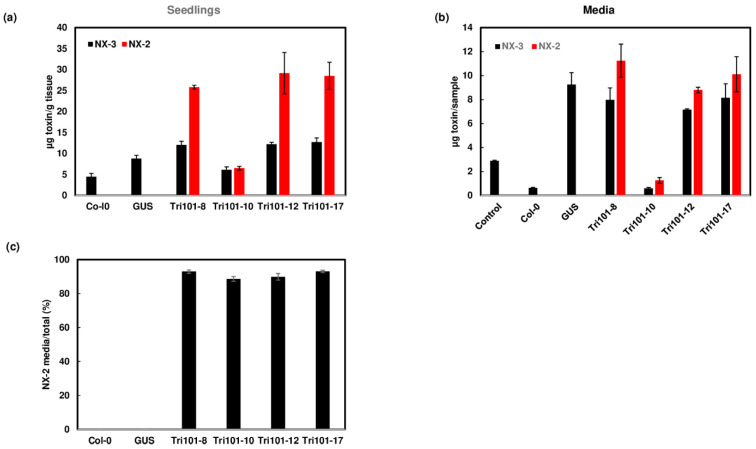
Conversion of NX-3 to NX-2 and excretion of NX-2 by *FgTri101* transgenic *Arabidopsis*. (**a**) NX-3 and NX-2 in the *Arabidopsis* seedlings after treatment with 50 mg/L NX-3 for 2 days. (**b**) NX-3 and NX-2 in the media after *Arabidopsis* seedlings were treated with 50 mg/L NX-3 for 2 days. Media without plant seedlings served as a control. Toxins in the media were measured in 1.4 mL aliquots from 20 mL of media and presented. (**c**) NX-2 in media in comparison to NX-3 in plant and media. Percentage of excreted NX-2: NX-2 media/ (NX-2 media + plant) × 100.

**Figure 7 toxins-13-00320-f007:**
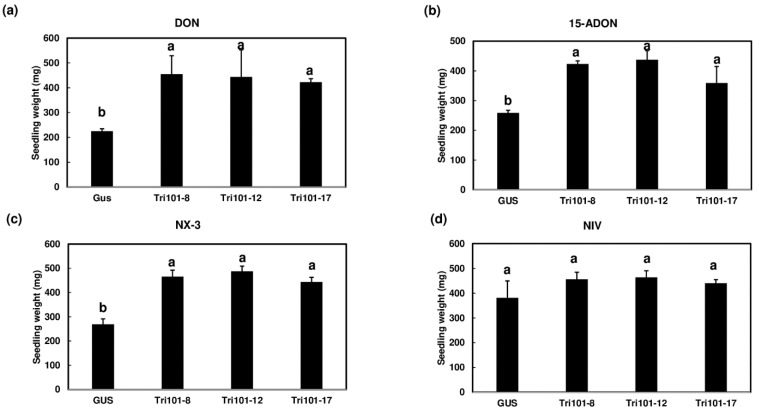
Weight comparison in transgenic *Arabidopsis* seedlings expressing *FgTri101* and GUS with trichothecenes treatments. *Arabidopsis* seedlings were treated with individual trichothecenes (50 mg/L) for 2 days: (**a**) DON; (**b**) 15-ADON; (**c**) NX-3; and (**d**) NIV. The data were analyzed by one-way ANOVA and Tukey’s post hoc test using JMP. Bars with different letters indicate statistically significant differences.

**Table 1 toxins-13-00320-t001:** Morphological comparison of transgenic *Arabidopsis* plants expressing *FgTri101* and GUS.

Genotypes	Flower Time (days)	Rosette Leaf Number ^a^	Plant Height (cm) ^b^	Shoot Number ^b^
GUS	27	5.4 ± 0.52	32.6 ± 2.29	5.6 ± 1.30
Tri101-8	27	5.1 ± 0.35	29.5 ± 2.09	6.5 ± 1.41
Tri101-12	27	4.6 * ± 0.50	31.5 ± 3.85	6.5 ± 1.69
Tri101-17	27	5.0 ± 0.53	31.5 ± 2.19	5.6 ± 1.19

^a^ The number of the rosette leaves was counted at 17 days after plants were transferred to soil. ^b^ The plant height and shoot were measured when plants matured at about 40 days grown in soil. * An asterisk (*) indicates significance at the 0.05 level by one-way ANOVA and Dunnett’s method, compared with GUS transgenic plants using JMP.

## Data Availability

Not applicable.

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
