# Peer review of "Detoxification and Excretion of Trichothecenes in Transgenic Arabidopsis thaliana Expressing Fusarium graminearum Trichothecene 3-O-acetyltransferase"

_toxins, 2021, doi:10.3390/toxins13050320_

Round 1
Reviewer 1 Report
Recommend publication.
Reviewer 2 Report
In this revised manuscript, the authors have made modifications based on my previous review report. Although there are cavities that remain to be further addressed, this article will provide insights into the field that the transgenic Arabidopsis expressing FgTRI101 can be used as a feasible approach to protect plant cells by detoxifying trichothecenes and excreting the acetylated toxins out of plant cells. This preliminary data will paving a way for further investigation of novel targets to reduce FHB and mycotoxin contamination in economically important plant crops.
This manuscript is a resubmission of an earlier submission. The following is a list of the peer review reports and author responses from that submission.
Round 1
Reviewer 1 Report
In this manuscript, the authors have attempted to describe the ability of F. graminearum trichothecene 3-O-acetyltransferase (FgTRI101) in detoxifying several trichothecenes produced by F. graminearum, including DON, 15-ADON, NX-3, and NIV by using the transgenic Arabidopsis plants expressing FgTRI101. In fact, using this research model, the authors showed some interesting observations that the FgTri101 transgenic plants acetylated DON to 3-ADON, 15-ADON to 3,15-diADON, and NX-3 to NX-2, and the majority of the converted toxins seem to be excreted into the media. However, there are major concerns arising from this preliminary data that fall into its rationale and scientific novelty.
Here are major comments:
1. Fine 61-66, the authors stated that there are studies that have done on TRI101 using wheat and barley expressing F. sporotrichioides TRI101 (FsTRI101). Thus, it’s essential that the authors should take into account the following points:
a. It’s not clear the rationale of performing the current study. Whether the current research is repeating or confirming the previous study?
b. The comparison should be conducted for in silico analysis of F. graminearum TRI101 vs F. sporotrichioides TRI101 as well as the capability of detoxifying trichothecenes in the same research model.
c. How’s about the resistance of transgenic plants expressing FgTri101 to Fusarium infection? The authors should consider adding additional data/experiments that use wheat and barley, not Arabidopsis, which are suitable for studying resistance plants against Fusarium infection.
2. The transgenic plant is a temporary FgTri101 expression in the plasmid, not integrated into the Arabidopsis genome? If so, how stable of the transgenic plant in its capability of detoxifying trichothecenes and excretion of modified trichothecenes into the media?
3. The mechanism of pump/secret acetylated trichothecenes to the outside of the cells is still ambiguous, and again, needs to be addressed using wheat and barley.
4. Important question lines on the unchanged NIV when applying this strategy on Arabidopsis. It is confusing and curious that if this is the same observation apply to wheat and barley.
5. Figure 1 needs to be re-compiled as a professional illustration to show the significant increase of FgTri101 expression in other transgenic plants, including FgTri101-8, -12, and -17.
Reviewer 2 Report
The manuscript reported that Arabidopsis plants transformed with the FgTRI101 gene were able to convert trichothecenes including DON, 15-ADON, and NX-3 into less toxic forms and to excrete the converted forms into the growth medium. This is an important finding and worth being published.
Among 3 selected transgenic lines, the expression of FgTRI101 at the mRNA level by the line Tri-101-12 was more than 700 fold higher than the lines Tri101-8 and 17, but the phenotypes of the 3 lines were not significantly different on the most measured parameters. Therefore, the expression of FgTRI101 at the protein level needs to be examined to get a whole pictures of the gene expression related to the phenotypes. The results of the protein work should be added and discussed in the manuscript.
In addition to a control line expressing GUS, the transgenic line Tri101-10 should be added as another control to examine the seedling growth and the toxin conversion and excretion.
Reviewer 3 Report
This paper reports the detoxification and excretion of trichothecenes in transgenic Aribidopsis thaliana. The paper is well written and easily understandable. The author have generated mutants by insertion of a trichothecene 3-O-acetyltransferase (FgTRI101) from F. graminearum into Arabidopsis thaliana by traditional agrobacteria mediated transformation. The mutants were grown in presence of DON, 15-ADON, NIV and NX-3 and it was shown that the mutants were able to acetylate DON, 15-ADON and NX-3 and secrete the toxins outside the plant cell.
It is a well-known approach to clone genes from fungi and transform them into plants in order to detoxify toxins, but the work presented here is well designed and the idea to use a 3-0-acetyltransferase from the fungi that produce the toxin is obviously a good idea. You could always argue that these kind of studies should have been done in monocotyledonous plants and not in a dicotyledons model plant. It could have been beneficial for the story, if the transporter was identified and knocked out in order to verify the full process of cytoplasmic acetylation and transport of the toxins. Never the less, I think that the story fits well to Toxins in the present form.
The paper seems to be in good shape and I only have minor comments:
Arabidopsis and not abidopsis in header for sup.material
Line 254-255 and 307-309 in the discussion are duplicated
Figure 1 and 2 in sup. Should be S1 and S2
Figure S2 and S3 could be more informative. Please provide masses that are measured and an explanation to “Sugars”
Could this approach also could be useful for other types of mycotoxins than only trichothecenes from fusaria? A small paragraph about relevance for other mycotoxins could be added to the discussion.